# Caught in a trap: DNA contamination in tsetse xenomonitoring can lead to over-estimates of *Trypanosoma brucei* infection

**Isabel Saldanha**[1]*, **Rachel Lea**[1], **Oliver Manangwa**[2], **Gala Garrod**[1], **Lee R. Haines**[3], **Álvaro Acosta-Serrano**[3], **Harriet Auty**[4], **Martha Betson**[5], **Jennifer S. Lord**[1], **Liam J. Morrison**[6], **Furaha Mramba**[2], **Stephen J. Torr**[1], **Lucas J. Cunningham**[7]

**1** Department of Vector Biology, Liverpool School of Tropical Medicine, Liverpool, United Kingdom, **2** Vector and Vector-borne Diseases Research Institute, Tanga, Tanzania, **3** Department of Biological Sciences, University of Notre Dame, Notre Dame, Indiana, United States of America, **4** School of Biodiversity, One Health & Veterinary Medicine, University of Glasgow, Glasgow, United Kingdom, **5** School of Veterinary Medicine, University of Surrey, Guildford, United Kingdom, **6** The Roslin Institute, Royal (Dick) School of Veterinary Studies, University of Edinburgh, Edinburgh, United Kingdom, **7** Department of Tropical Disease Biology, Liverpool School of Tropical Medicine, Liverpool, United Kingdom

* isabel.saldanha@lstmed.ac.uk

## Abstract

### Background

Tsetse flies (*Glossina* sp.) are vectors of *Trypanosoma brucei* subspecies that cause human African trypanosomiasis (HAT). Capturing and screening tsetse is critical for HAT surveillance. Classically, tsetse have been microscopically analysed to identify trypano-somes, but this is increasingly replaced with molecular xenomonitoring. Nonetheless, sensitive *T. brucei*-detection assays, such as TBR-PCR, are vulnerable to DNA cross-contamination. This may occur at capture, when often multiple live tsetse are retained temporarily in the cage of a trap. This study set out to determine whether infected tsetse can contaminate naïve tsetse with *T. brucei* DNA via faeces when co-housed.

### Methodology/Principle findings

Insectary-reared teneral *G. morsitans morsitans* were fed an infectious *T. b. brucei*-spiked bloodmeal. At 19 days post-infection, infected and naïve tsetse were caged together in the following ratios: (T1) 9:3, (T2) 6:6 (T3) 1:11 and a control (C0) 0:12 in triplicate. Following 24-hour incubation, DNA was extracted from each fly and screened for parasite DNA presence using PCR and qPCR. All insectary-reared infected flies were positive for *T. brucei* DNA using TBR-qPCR. However, naïve tsetse also tested positive. Even at a ratio of 1 infected to 11 naïve flies, 91% of naïve tsetse gave positive TBR-qPCR results. Furthermore, the quantity of *T. brucei* DNA detected in naïve tsetse was significantly correlated with cage infection ratio. With evidence of cross-contamination, field-caught tsetse from Tanzania were then assessed using the same screening protocol. End-point TBR-PCR predicted a sample population prevalence of 24.8%. Using qPCR and Cq cut-offs optimised on

**Data Availability Statement:** All data generated during this project is available at DOIs https://doi.org/10.6084/m9.figshare.25298644.v1 and https://doi.org/10.6084/m9.figshare.25298689.v1.

**Funding:** Funding was provided by Biotechnology and Biological Sciences Research Council (BBSRC; BB/S01375X/1 (to HA, JSL, LJM, FM, SJT); www.ukri.org/councils/bbsrc/) and Bill & Melinda Gates Foundation (INV-031337, INV-001785, INV-046509 (to SJT); www.gatesfoundation.org/). The Roslin Institute is supported by core funding from the BBSRC (BBS/E/D/20002173, BBS/E/RL/230002C; www.ukri.org/councils/bbsrc/). The funders had no role in study design, data collection and analysis, decision to publish, or preparation of the manuscript.

**Competing interests:** The authors have declared that no competing interests exist.

insectary-reared flies, we estimated that prevalence was 0.5% (95% confidence interval [0.36, 0.73]).

## Conclusions/Significance

Our results show that infected tsetse can contaminate naïve flies with *T. brucei* DNA when co-caged, and that the level of contamination can be extensive. Whilst simple PCR may overestimate infection prevalence, quantitative PCR offers a means of eliminating false positives.

## Author summary

Tsetse flies (*Glossina* sp.) are vectors of *Trypanosoma brucei* parasites that cause human African trypanosomiasis, also known as sleeping sickness. As part of disease surveillance, tsetse can be captured in traps and checked for parasite presence. The molecular screening of disease vectors (such as mosquitoes, ticks and blackflies) for the presence of pathogen DNA has gained popularity in recent years. However, DNA contamination may occur at capture when live vectors are retained for a limited period in a trap cage. To explore this, we conducted experiments, initially with laboratory-reared tsetse and then field-caught tsetse from Tanzania. Our results show that infected tsetse can contaminate uninfected tsetse with *T. brucei* DNA when retained together in a trap cage, and that the level of contamination can be extensive. Infected tsetse consistently shed *T. brucei* DNA in their faeces, which in turn contaminates other tsetse. This can produce false-positive results, leading to inaccurate reporting of infection prevalence. These findings impact not only trypanosomiasis surveillance, but may also have ramifications for the xenomonitoring of other vector-borne neglected diseases. Future work should explore whether pathogen DNA contamination routes exist in other vector species and, if so, the methods to mitigate DNA contamination in entomological traps.

## Background

Tsetse flies (*Glossina* sp.) are the primary vector for several species of *Trypanosoma* which cause the neglected tropical disease human African trypanosomiasis (HAT) as well as animal African trypanosomiasis (AAT) [1]. The sub-genera *Trypanozoon* comprises three closely related species: *T. brucei* and the animal pathogens *T. b. evansi* and *T. b. equiperdum*. A species of both human and animal clinical significance, *T. brucei* can be further divided into three sub-species: *T. brucei rhodesiense* is the zoonotic cause of East African 'Rhodesian' HAT (rHAT) and can also cause AAT, *T. brucei gambiense*, is anthroponotic, causing West African 'Gambian' HAT (gHAT) and *T. brucei brucei* causes AAT in livestock across sub-Saharan Africa.

Collecting and screening tsetse for the presence of *T. brucei* is a HAT surveillance technique with a long history, having been standardised in 1924 by Lloyd and Johnson [2]. Systematic sampling of tsetse populations allows not only the monitoring of tsetse population dynamics, but also parasite prevalence within a particular environment. The presence of HAT pathogens in tsetse populations is considered an aspect of 'tsetse challenge', an important part of calculating HAT transmission risk [3,4]. Historically, individual tsetse have been collected, dissected and subjected to microscopic analysis to determine whether *Trypanosoma* sp. were present

and to identify the subspecies depending on which fly tissues were colonised [2]. This technique was the gold standard for identification of trypanosome infection in tsetse for several decades, and is still in use today as the only way to positively identify an active infection [1,2,5]. However, this method is labour-intensive and suffers from poor sensitivity and specificity due to limitations in microscope resolution, similarities in *Trypanosoma* physical morphology and the inability to designate maturity of infection stage within the fly [1,5–7].

Over the last decade, molecular xenomonitoring has largely replaced traditional microscopy detection of parasites. This is where hematophagous insect vectors are screened for genetic targets indicative of pathogen presence, as a proxy for human or animal disease occurrence. Xenomonitoring has been developed for a range of arthropod vector-borne diseases, including HAT, AAT, lymphatic filariasis, onchocerciasis, Dengue, Chikungunya and Zika [8–14]. The benefits of molecular xenomonitoring include the potential for high-throughput sample analysis and very high sensitivity and specificity, with estimates of 1.9–9.3 times greater sensitivity than dissection [5,15].

A variety of molecular assays using a range of *T. brucei* genomic targets have been developed for xenomonitoring purposes. Minichromosome satellite DNA tandem repeat regions are the most sensitive targets, with copy numbers estimated at 10,000 in *T. brucei* sensu-lato [16]. Although this 177-bp *T. brucei* s-l repeat (TBR) region was recently confirmed to be more heterogeneous than initially anticipated [17], it remains the most sensitive and widely-used molecular target in the form of TBR-PCR, SYBR green TBR-qPCR and a novel probe-based TBR-qPCR assay [17–19].

However, such highly sensitive methods can lead to problems in determining a true biological infection within the vector. Xenomonitoring can be a powerful disease ecology tool, in being able to detect parasite presence within a given environment with a high degree of sensitivity. Yet it is also used to estimate trypanosome prevalence. The mere presence of target DNA within a sample is usually interpreted as a 'positive' fly. However, it is impossible to determine a true mature parasite infection, with a viable transmission risk, from an immature infection or from a passing infected bloodmeal. The results may be particularly difficult to interpret when an end-point assay is used (PCR, LAMP, RPA) as opposed to quantitative DNA methods (qPCR). An end-point assay can only indicate the presence or absence of pathogen DNA, yet PCR results are often reported as sample population infection rate or prevalence. Sensitive DNA amplification methods are also susceptible to DNA contamination [20].

Contamination with parasite DNA can occur at several stages in the xenomonitoring process: (i) molecular screening, (ii) DNA extraction or (iii) when flies are trapped and collected [21]. Whilst inclusion of controls can easily eliminate contamination at the screening and DNA extraction stages, contamination at the trapping phase is not possible to determine retrospectively.

Several studies that have used TBR-PCR to screen tsetse flies have reported a higher-than-expected proportion of flies testing positive for *T. brucei* s-l DNA. Whereas a *T. brucei* s-l infection prevalence of <1% might be expected in wild fly populations [22], studies using TBR-PCR have reported far higher proportions. From 8.9% (63/706) [23], 13.7% [24] and 15% [25], to more than 40% [26] and up to 70.7% [27]. In a study reporting *T. brucei* s-l infections in 46% of midgut-positive flies, McNamara *et al* discussed the possibility of false-positive TBR-PCR due to trace *T. brucei* DNA residue from previous bloodmeal(s) [28]. At the time, this was countered with evidence of rapid degradation of *Trypanosoma* DNA in the midgut following an infectious bloodmeal [28]. However more recent evidence has shown that *T. b. brucei* DNA can remain detectable in the midgut of an uninfected or refractory tsetse for up to six days post-feed [29].

Tsetse traps currently in widespread use were designed before the rise of molecular methods, and whilst the trypanosome detection methods have changed, the trapping and collection

methods have largely remained the same. For a cloth trap such as Nzi, blue and black panels paired with transparent mesh netting attract and direct tsetse into a trap cage where they are held until collection [30]. The trap cage may be a mesh bag or, more commonly, a transparent plastic bottle. Typically set for 24–48 hours, tsetse traps may capture anywhere from zero to several hundred tsetse, dependant on location and local population density. Agitated tsetse defecate or excrete larger (wet) volume of waste products (such as faeces) under heat stress or high humidity [31], which in turn forms the basis for a DNA contamination pathway.

Tsetse faeces, also known as frass, are composed of digested bloodmeal excreta. In an infected tsetse, faeces can also contain *T. brucei* DNA from lysed or digested parasites. Previous studies have shown that experimentally-infected tsetse flies excrete *T. brucei* DNA in excreta or faeces and that this is detectable by PCR [32,33]. This provides a potential route of *T. brucei* DNA contamination within a tsetse trap. Due to their size and energetic needs, tsetse take relatively large bloodmeals, with the bloodmeals taken by *G. m. morsitans* and *G. pallidipes* ranging between 37.3–62.3 mg and 53.9–76.3 mg of wet mass [34]. Although much of this is metabolised, it has been estimated that for every 1 mg of blood (dry weight) ingested, a tsetse will excrete approximately 0.5 mg [35].

In this study, we tested the hypothesis that trypanosome-infected tsetse flies can contaminate uninfected individuals with *T. brucei* DNA within a trap environment, subsequently leading to biased estimates of trypanosome infection when screening trap-caught tsetse using TBR-target molecular methods. To test this hypothesis, we conducted laboratory-based studies to assess whether mixing infected and uninfected tsetse within a cage resulted in both groups of flies being positive for TBR-PCR. We also examined wild-caught tsetse to assess whether there was evidence of cross-contamination occurring in practice. Finally, we developed a means of estimating infection prevalence accurately in settings where contamination is suspected or known to have occurred.

## Methods

### Experimental infection of tsetse flies

A total of 140 (80 male, 60 female) teneral *Glossina morsitans morsitans* aged 12–48 hours post-emergence were fed a defibrinated equine bloodmeal (TCS Biosciences Ltd, UK) containing approximately $1\times10^6$ per mL of bloodstream form *T. brucei brucei* (strain TSW196 [36]) in SAPO containment facilities at Liverpool School of Tropical Medicine (LSTM). After 24 hours, flies containing a visible bloodmeal in their abdomens (n = 110; 51 female, 59 male) were selected and placed into solitary cells (S1 Fig). Fed flies were maintained for 19 days post-infection by blood-feeding every 2–3 days in a temperature- (25 ± 2°C) and humidity-controlled (68%–79%) environment. Individual fly faecal samples were collected by placing 25mm filter paper discs (Whatman, UK) underneath each fly cell (S1 Fig). Faecal samples were collected at the following intervals: 6–7 days (n = 45), 8–9 days (n = 45), 10–12 days (n = 110) and 13–14 days (n = 110) post-infectious bloodmeal. Faecal samples were stored in individual microcentrifuge tubes at room temperature (RT) until further processing. Of 110 flies that consumed an infectious bloodmeal, 106 (50 female, 56 male) survived to 19 days post-infection.

### Trap experiments

TBR-qPCR screening of tsetse faecal samples collected 10–14 days post-infection was used to determine individual fly infection status [32]. This time was chosen as it surpassed the seven-day period where dead *T. brucei* DNA from an infectious bloodmeal would have remained detectable [29]. At 19 days post-infection, after 72 hours starvation to mimic field conditions

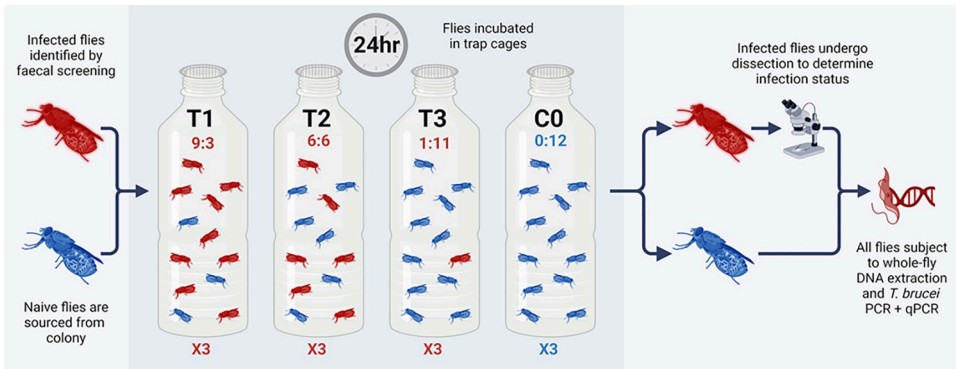

**Fig 1. A flow diagram depicting basic experimental framework for the trap experiments.** Figure created using biorender.com (www.biorender.com [accessed 01/02/24]).

where tsetse would be seeking a host, 48 trypanosome-infected flies (IFs) with intact wings were selected and marked with a unique identifier (artist's oil paint [Windsor and Newton, UK] applied to the dorsal surface of the thorax; S1 Fig). Remaining flies (n = 62, a mixture of refractory and infected) remained in solitary cells. Forceps were cleaned with 10% bleach and rinsed in nuclease-free water between handling of each fly. IFs and 96 uninfected (naïve) flies (UFs) were placed in plastic bottles similar to the cages used for trapping, namely, 250mL transparent plastic bottles with a fine mesh cover in place of lid (S1 Fig). This experimental design gave a density of 48 flies per litre, mimicking field catches [10]. The numbers of IFs and UFs in the bottles was varied according to three classes of treatment and a control (Fig 1). The three treatments comprised IF:UF in ratios of: (T1) 9:3, (T2) 6:6, (T3) 1:11 and control (C0) 0:12. T3 represents the low infection ratio most likely to be encountered in the field [22]. Fly sex ratios were balanced where possible (S1 Table). Each treatment was replicated three times (A, B and C). To test for localised airborne DNA contamination, control traps C0-A and C0-B were placed within close proximity (<1 metre) to treatment traps (T1-T3), whereas C0-C was placed in a separate room. Once flies had been placed into trap vessels and had sufficient time to revive (approximately 30 minutes), they were incubated for 24 hours in temperature- and humidity-controlled conditions (Fig 1). Once complete, all tsetse were sedated in a cold room at 5–10˚C. UFs were placed into individual collection tubes containing chilled 100% ethanol and subsequently stored at room temperature (RT). All IFs (n = 48) and a proportion of left-over flies (n = 23) were stored in individual tubes on ice for immediate dissection.

## Tsetse dissection and microscopic analysis

To confirm infection status, all IFs (n = 48) and some remaining (fed infectious bloodmeal but not infected) flies (n = 23) were dissected and inspected by light microscopy at 400X magnification to detect trypanosome infection as described elsewhere [2]. Visible procyclic trypomastigote forms in the midgut (MG) were recorded as infection-positive. Salivary glands (SGs) were not inspected for presence of epimastigote or metacyclic trypomastigote forms as SG infection is only visible after ~21 days and faecal screening is thought to only be indicative of midgut infection status [32]. It is worth noting that at 20 days, no bloodstream forms from initial *T. brucei* infectious bloodmeal would have remained within the tsetse. Dissection equipment was cleansed with 10% bleach and rinsed in nuclease-free water between each sample. A new glass slide was used for each fly. Once dissection was complete, each individual fly was placed into collection tube containing chilled 100% ethanol and stored at RT.

### Field sampling and collection of tsetse

As part of the BBSRC-funded study ENABLES (BB/S01375X/1) and under the auspices of the Tanzania Commission for Science and Technology (COSTECH; permit codes 2019-414- NA-2018- 360 and 2019-413- NA-2018- 360), sampling of tsetse species *G. pallidipes*, *G. swynnertoni* and *G. morsitans* took place at sites in Tarangire National Park and Simanjiro district, Tanzania, in August 2019. The Tarangire National Park covers 2,850 km$^2$ and is bordered by Simanjiro, Babati and Monduli districts [37]. The altitude varies between 1356 m and 1605 m, rising from southeast to northwest on a raised plateau. The vegetation can be split into seven main types: grassland and floodplains; *Acacia tortilis* parkland; tall *Acacia* woodland; drainage line woodland; *Acacia-Commiphora* woodland; *Combretum-Dalbergia* woodland; and rocky hills [38]. In August 2019, 51 Nzi traps were set within the Tarangire National Park (transects TA and TB) and 38 outside and to the east in Simanjiro District (transects BA and BB). Location coordinates for each trap are listed in S2 Table. Trapping was carried out as described previously [10]. In short, Nzi traps [30] baited with acetone (100 mg/h), 1-octen-3-ol (1 mg/h), 4-methylphenol (0.5 mg/h) and 3-n-propyphenol (0.1 mg/h) [39,40] were deployed for 72 h and flies collected every 24 h (Fig 2). Trapped flies were held in-situ in a trap cage (1000 mL plastic bottle) for approximately 24 hours until collection. The species and sex of individual tsetse were recorded, each fly was assigned an ID number and stored individually in 1.5 mL collection tubes containing ~1mL of 100% ethanol. All flies were deceased upon collection. Although sampling was carried out for the primary purpose of population abundance monitoring and modelling, the opportunity was taken to collect a proportion of the trapped flies for molecular xenomonitoring purposes. Due to high catch numbers at some sites (>500 tsetse/

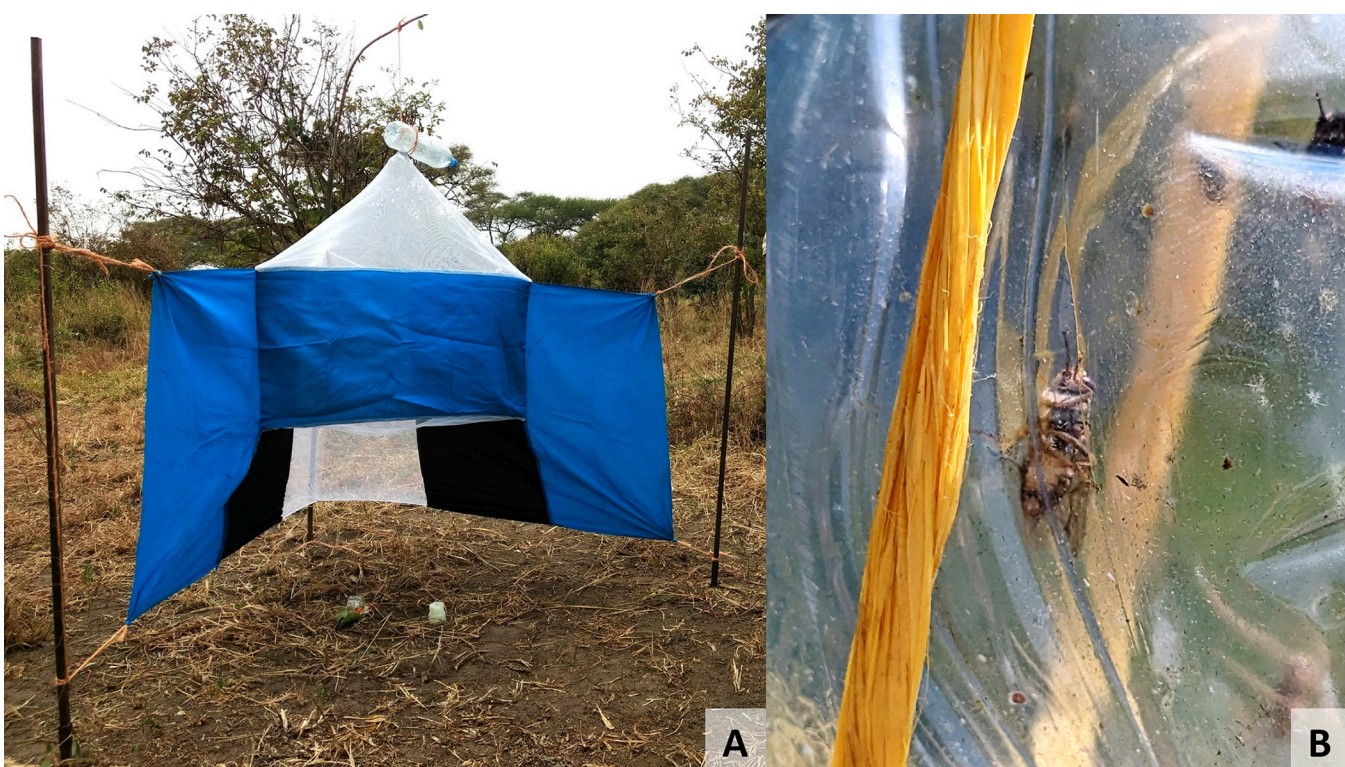

**Fig 2.** An example of an Nzi trap with trap cage at the apex (A) and detail of a *Glossina* sp. captured within the trap cage (B).

trap/day), not all flies that were trapped were collected and screened. Flies were selected randomly for collection.

## DNA extraction

For faecal samples collected from insectary-reared tsetse (S1 Fig), a 2 mm Harris micro-punch was used to extract a single faecal sample from each filter paper. Hole punch and forceps were cleaned with 10% bleach and then nuclease-free water between each sample. The samples of filter paper were placed into individual collection tubes containing 40 μL sterile phosphate-buffered saline (PBS) and incubated at 37˚C for 1 hour [41] on a rocker set at 5 oscillations per minute. DNA was extracted and purified from the disc and PBS using a DNeasy 96 Blood and Tissue Kit (QIAGEN, Hilden, Germany) following the manufacturer's protocol for purification of DNA from animal tissues. Eventual purified DNA was eluted in 80 μL of elution buffer AE.

For tsetse flies (both experiment and field), whole intact tsetse or total dissected remains were placed into individual collection tubes and incubated at 56˚C for 3 hours on a rocker set at 5 oscillations per minute to remove ethanol. DNA was extracted and purified using a DNeasy 96 Blood and Tissue Kit (QIAGEN, Hilden, Germany) following the manufacturer's protocol, slightly optimised for large insect processing with the addition of a mechanical lysis step. In short, after ethanol removal, a quarter-inch diameter stainless-steel ball (Dejay Distribution Ltd, UK) was placed into each tube. After adding Buffer-ATL/Proteinase K, samples were then mechanically lysed at 15 Hz for 20 seconds for two rounds using a TissueLyser II (QIAGEN, Hilden, Germany). Following centrifugation at 2000 xg for 1 minute, samples were incubated at 56˚C for 14 hours. Eventual purified DNA was eluted in 80μL elution buffer AE.

For insectary-reared flies, a negative extraction control (NEC) was included every 3–18 flies (26 NEC to 206 flies total). For field flies, an NEC was included for every 93 flies (32 NEC to 2777 flies total).

## TBR-PCR

PCR primers used in the study are detailed in Table 1. TBR-PCR reactions were carried out using MyTaq Red Mix (Meridian Bioscience, Cincinnati, US) following the manufacturer's protocol. In brief, 5 μL of DNA template was added to 12.5 μL 2X MyTaq Red Mix, 0.5 μL of each 10 μM forward and reverse primer and 6.5 μL nuclease-free water to give a 25 μL total reaction volume. For TBR-PCR reactions to generate amplified products for sequencing, all

**Table 1. Trypanosome detection primers used in the study.**

| Oligo Name | Sequence (5' ➔ 3') | Target | Assay Name | Source |
|---|---|---|---|---|
| TBR_PCR_F | CGAATGAATATTAAACAATGCGCAGT | *Trypanozoon* minichromosome satellite DNA repeat | TBR-PCR | [19] |
| TBR_PCR_R | AGAACCATTTATTAGCTTTGTTGC | | | |
| TBR_QPCR_F | CGCAGTTAACGCTATTATACACA | *Trypanozoon* minichromosome satellite DNA repeat | TBR-qPCR | [42] |
| TBR_QPCR_R | CATTAAACACTAAAGAACAGCGT | | | |
| TBR_QPCR_PRB | FAM-TGTGCAACATTAAATACAAGTGTGT-ZEN | | | |
| PLC1 | CAGTGTTGCGCTTAAATCCA | *Trypanozoon* glycosylphosphatidylinositol-specific phospholipase-C gene | PLC-qPCR / HAT-HRM | [9,43] |
| PLC2 | CCCGCCAATACTGACATCTT | | | |
| TbRh1 | GAAGCGGAAGCAAGAATGAC | *Serum resistance-associated protein* gene | HAT-HRM | [9] |
| TbRh2 | GGCGCAAGACTTGTAAGAGC | | | |
| TgsGP1 | CGAAGAACAAAGCCGTAGCG | *T. b. gambiense-specific glycoprotein* gene | HAT-HRM | [9] |
| TgsGP2 | CCGTTCCCGCTTCTACTACC | | | |

reagent volumes were doubled to give a total reaction volume of 50 μL (10 μL template DNA). Thermocycling conditions for TBR-PCR were as follows; 3 minutes at 95°C initial denaturation, followed by 35 cycles of 15 seconds denaturation at 95°C, 15 seconds annealing at 55°C, and 20 seconds extension at 72°C, followed by final extension for 2 minutes at 72°C. Thermocycling was carried out using an Applied Biosystems Veriti thermal cycler (Life Technologies, Carlsbad, US). PCR products were separated by agarose gel electrophoresis and visualised using a gel documentation system (Syngene International, India; S2 Fig). *Trypanosoma brucei* M249 DNA at concentration of 1 ng/μL was used as positive template control (PTC) for TBR-PCR assays. Nuclease-free water was used as no-template control (NTC) for all assays. All pre-amplification set-up was carried out in a STARLAB AirClean 600 workstation (STARLAB, UK) in a separate room to post-amplification analysis.

## TBR-qPCR and PLC-qPCR

Quantitative PCR (qPCR) primers used in the study are detailed in Table 1. TBR-qPCR reactions were carried out using Bio-Rad SsoAdvanced Universal Probes Supermix (Bio-Rad Laboratories, Hercules, US) in line with the manufacturer's protocol. In short, 5 μL template DNA was mixed with 10 μL SsoAdvanced Universal Probes Supermix (2X), 0.4 μM forward and reverse primers, 0.2 μM probe and nuclease-free water added to a 20 μL total reaction volume. Thermal cycling conditions were as follows; initial denaturation at 95°C for 3 minutes followed by 40 cycles of denaturation at 95°C for 10 seconds and annealing and extension at 59°C for 12 seconds. Data was captured during the annealing and extension step. Thermocycling, fluorescence detection and data capture was carried out using a Mic and micPCR v.2.9.0 software (Bio Molecular Systems, Upper Coomera, Australia).

PLC-qPCR screening of insectary-reared and field collected flies was performed using Agilent Brilliant III Ultra-Fast Master Mix (Agilent Technologies, Santa Clara, USA) following the manufacturer's protocol. Briefly, 5 μL of template DNA was mixed with 10 μL Ultra-Fast Master Mix (2X), 200 nM of forward and reverse and primer and nuclease-free water to a total reaction volume of 20 μL. Thermal cycling conditions were as follows: initial denaturation at 95°C for 3 minutes followed by 40 cycles of denaturation at 95°C for 10 seconds and annealing and extension at 60°C for 20 seconds. Data was captured during the annealing and extension step. Following cycling, a melt step was performed between 65–95°C at 0.3°C per second. Thermocycling, fluorescence detection and data capture was carried out using a Mic and micPCR v.2.9.0 software (Bio Molecular Systems, Upper Coomera, Australia).

Additional PLC-qPCR screening in field flies was carried out as part of a multiplex HAT-HRM assay using reaction conditions and thermocycling as described previously [9]. A positive PLC-qPCR sample was defined as a sample with a single melt peak that occurred at 79.1°C and crossed a baseline threshold of 10% of the maximum normalized fluorescence (dF/dT) of the highest peak. A positive *T. b. rhodesiense* sample was defined as a sample with melt peaks that occurred at both 79.1°C and 84.2°C and crossed a baseline threshold of 10% of the maximum normalized fluorescence (dF/dT) of the highest peak.

*Trypanosoma brucei* M249 DNA at concentration of 1 ng/μL was used as positive template control (PTC) for the TBR-qPCR and PLC-qPCR assays. Nuclease-free water was used as NTC for all assays. All pre-amplification set-up was carried out in a STARLAB AirClean 600 workstation (STARLAB, UK) in a separate room to post-amplification analysis.

## PCR product sequencing

To confirm amplification of target *T. brucei* DNA in field samples, TBR-PCR products from a sub-sample of previously confirmed TBR-PCR positive field flies (n = 93/688) were purified

and sequenced. The 173 bp TBR-PCR target products were excised and purified using an Exo-CIP Rapid PCR Cleanup Kit (New England Biolabs, Ipswich, USA) following the manufacturer's protocol. Resultant purified DNA was eluted in 20 µL elution buffer. Sanger sequencing was performed by Source BioScience (Source BioScience Limited, Nottingham, UK) using both TBR_PCR_F and TBR_PCR_R primers (Table 1). Sequence clean-up and alignments were performed in BioEdit v7.2 [44]. Resultant sequences were subject to BLAST nucleotide analysis (National Centre for Biotechnology Information) against the target *T. brucei* satellite DNA entry (accession number K00392.1).

## Statistical analyses

All data were collated into a centralised database in Excel (Microsoft). Further analyses and data visualisation were performed using GraphPad Prism v10. All data are presented as the mean ± standard error (SE). For fly experiment results, Pearson's correlation coefficient was used to determine if there was an association between proportion of IFs (trap treatment) with UF TBR-qPCR Cq values. One-way ANOVA was used to determine if there were statistically significant differences in mean TBR-qPCR Cq values obtained from UFs in T1, T2 and T3. Student's T-test (2-tailed) was used to determine if there was a statistically significant difference between mean Cq values obtained from screening IF and UF whole-fly DNA. Mann-Whitney U Test was used to test if there was a significant difference between TBR-qPCR Cq values from male and female flies.

# Results

## Detection of *T. brucei* DNA in insectary-reared, experimental flies

Screening by TBR-qPCR revealed that flies hosting a trypanosome infection (IFs) produced Cq values between 14.46–21.57 (mean = 17.74, ±0.108 SE), which indicates a high quantity of TBR target DNA in infected flies (Fig 3A). However naïve uninfected flies (UFs), when co-housed with infected ones for 24 hours, were also positive for TBR target DNA. There was a strong negative correlation between UF TBR-qPCR Cq value and proportion of IFs in the trap ($r[68] = -0.8153$, $p = <0.0001$; Fig 3A). In other words, the quantity of DNA contamination was proportional to the infection rate of the trap. There were also distinct differences in the TBR-qPCR Cq values for UFs across the three different infection ratio treatments ($F[2, 17] = 40.80$, $p = <0.0001$; Fig 3A). Multiple comparison tests confirmed significant differences between all treatments; T1 UF and T2 UF (-3.118 mean Cq, $p = 0.0094$), T1 UF and T3 UF (-6.983 mean Cq, $p = <0.0001$) and T2 UF and T3 UF (-3.865 mean Cq, $p = <0.0001$). End-point TBR-PCR screening produced similar results to TBR-qPCR screening (S2 Fig), with amplification recorded in 100% of IFs, 100% of T1 UFs (n = 9/9), 100% of T2 UFs (n = 18/18) and 69% of T3 UFs (n = 23/33).

Contamination was evident even with an assay with single-copy target and lower sensitivity (PLC-qPCR), albeit to a lesser extent (Fig 3B). Only 9.1% of T3 UFs recorded amplification by PLC-qPCR, compared to 90.9% by TBR-qPCR (Fig 3).

Contamination was detected in control bottles placed within close proximity (< 1 metre) to bottles containing infected tsetse (C0-A, C0-B), but not in a control bottle placed in separate room (C0-C). Low-level amplification (Cq >35) was detected in 28.6% of UFs by TBR-PCR and 50% of UFs by TBR-qPCR across C0-A and C0-B with mean Cq 35.23 ±0.285 SE. UFs in C0-C (placed in a separate room) recorded no amplification by TBR-PCR, TBR-qPCR (Fig 3A).

The DNA contamination evidenced in the results did not occur at either the DNA extraction or amplification stages. Of 26 total extraction controls (NEC), zero recorded amplification

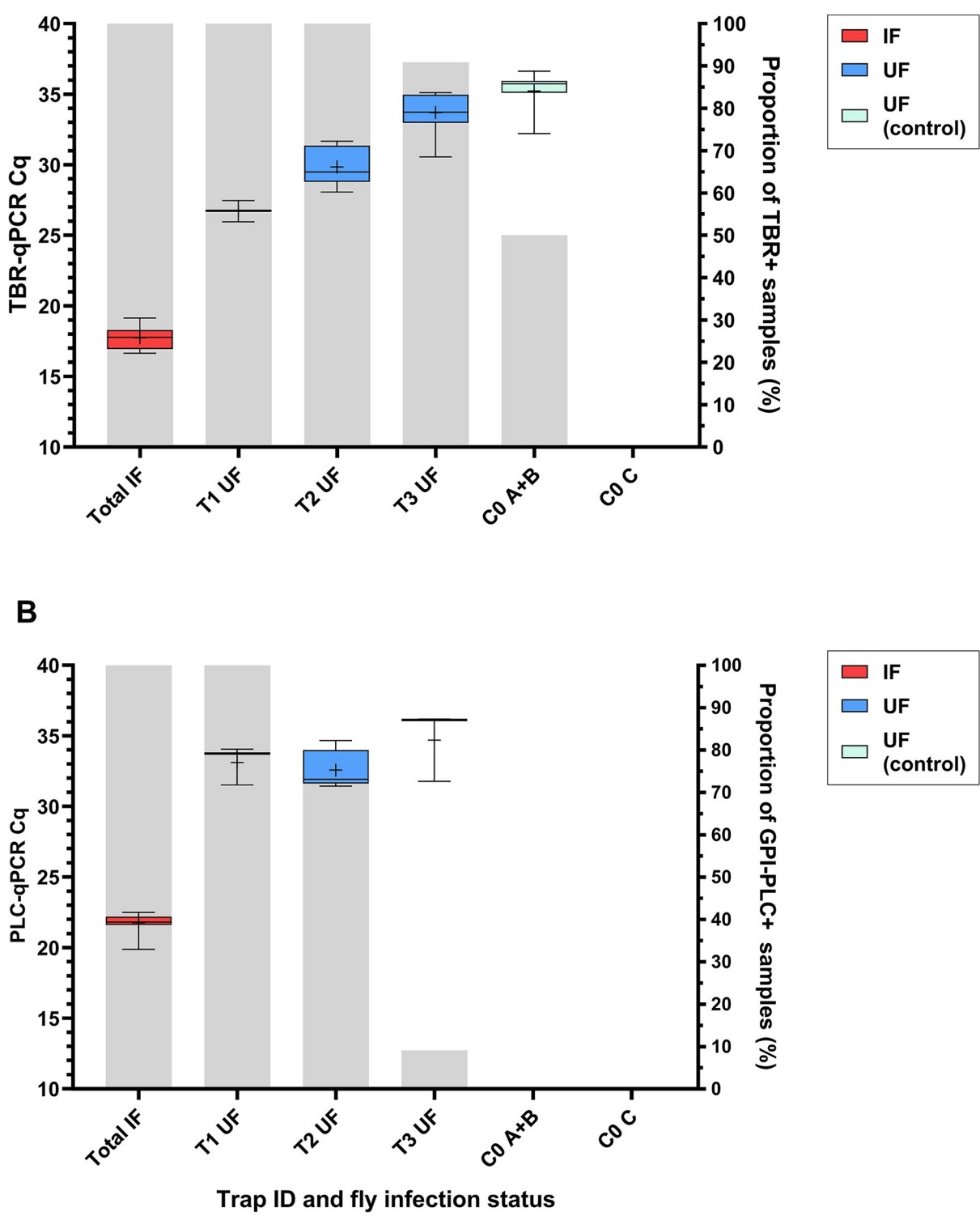

**Fig 3.** Box-and-whisker plots showing Cq value data from *T. brucei* (A) multi-copy target TBR-qPCR screening and (B) single-copy target PLC-qPCR screening of infected flies (IF) and naïve (UF) across four trap types (T1-T3, C0). C0 A+B were placed within close proximity (< 1 metre) of experiments (T1-3), C0 C was placed in a separate room. This was to test localised airborne DNA contamination. Crosses represent the mean Cq values. Grey bars display proportion of samples recording amplification using respective qPCR assays.

**Table 2. A table detailing sex, transect and TBR-PCR positive results breakdown of field-caught tsetse by species (*Glossina* sp.).**

| Species | Total | Sex | | Transect | | | | TBR-PCR+ | |
|---|---|---|---|---|---|---|---|---|---|
| | | Male | Female | TA | TB | BA | BB | Freq. | PCR+ prop. |
| *G. pallidipes* | 1675 | 553 | 1122 | 814 | 0 | 860 | 1 | 666 | 39.76% |
| *G. swynnertoni* | 1053 | 468 | 585 | 354 | 696 | 3 | 0 | 18 | 1.71% |
| *G. m. morsitans* | 49 | 17 | 32 | 49 | 0 | 0 | 0 | 4 | 8.16% |
| **All species** | **2777** | **1038** | **1739** | **1217** | **696** | **863** | **1** | **688** | **24.77%** |

Transects TA and TB consist of traps within Tarangire National Park. Transects BA and BB consist of traps in Simanjiro District close to the border of Tarangire National Park. 'Freq.' represents frequency. 'PCR+ prop.' is number/proportion of tsetse samples that produced diagnostic 173-bp TBR-PCR product.

by TBR-PCR or PLC-qPCR. However, one NEC did produce amplification by TBR-qPCR (Cq 34.29). It should be noted that this particular NEC was situated between IF samples containing high concentration of *T. brucei* DNA (Cq < 20). The fact that 13 other NECs in this extraction did not record amplification by any assay suggests that this was localised cross-contamination that did not affect other samples in the extraction. Of all NTCs across TBR-PCR (n = 4), TBR-qPCR (n = 7) and PLC-qPCR (n = 7), none produced amplification regardless of assay.

## Tsetse faecal screening as a predictor of infection status

Experimentally-infected tsetse excrete *T. brucei* DNA in their faeces, and screening these faeces can determine tsetse midgut infection with high accuracy (S3 Fig). Microscopy revealed that 100% (n = 48) of IFs selected for experiments, based on faecal screening, had developed mature midgut infection by 20 days post-infection.

## Detection of *T. brucei* DNA in field-collected flies

A total of 2777 tsetse were collected from traps in Tanzania (Table 2). TBR-PCR was performed on all 2777 flies, of which 688 (24.77%) tested positive. Of these, 661 samples had adequate DNA remaining and were subsequently screened using TBR-qPCR, of which 640 recorded amplification (Cq < 40). The amount of *T. brucei* DNA detected in samples varied more widely than in experimental flies, with TBR-qPCR Cq values from 4.59 to 38.52 and mean of 27.19 ±0.170 SE (Fig 4). There was no significant difference in median TBR-qPCR Cq values from females (median = 26.22) and males (median = 25.97, *p* = 0.5336; Fig 4C). No *T. b. rhodesiense* DNA was detected by HAT-HRM in any of the samples. Across all catches (n = 62), catch size varied widely from 1 to 420, with mean catch size of 89.35 ±12.494 SE (S4 Fig). Therefore, fly density within the traps varied from 1 to 420 flies per litre, with mean density of 89.35 (±12.494 SE) flies per litre and median density of 42 flies per litre.

DNA contamination was ruled out at both the DNA extraction and amplification stages as none of the NECs screened by TBR-PCR (n = 32) had amplification. However, of nine NECs screened by TBR-qPCR, two recorded low-level amplification (Cq 36.72, 34.54). In both cases, NECs were surrounded by samples containing high quantity of *T. brucei* DNA (Cq < 30) during plate DNA extraction. Therefore, these were considered to be instances of localised cross-contamination. Across NTCs screened by TBR-PCR (n = 32), TBR-qPCR (n = 23) and PLC-qPCR (n = 1), none recorded amplification.

## Estimation of sample population *T. brucei* infection prevalence

Of the total number of *T. brucei* positive field caught tsetse (*n* = 688/2777), 26 lacked sufficient volume of template and so were not included in the rest of the study. Calculating the sample

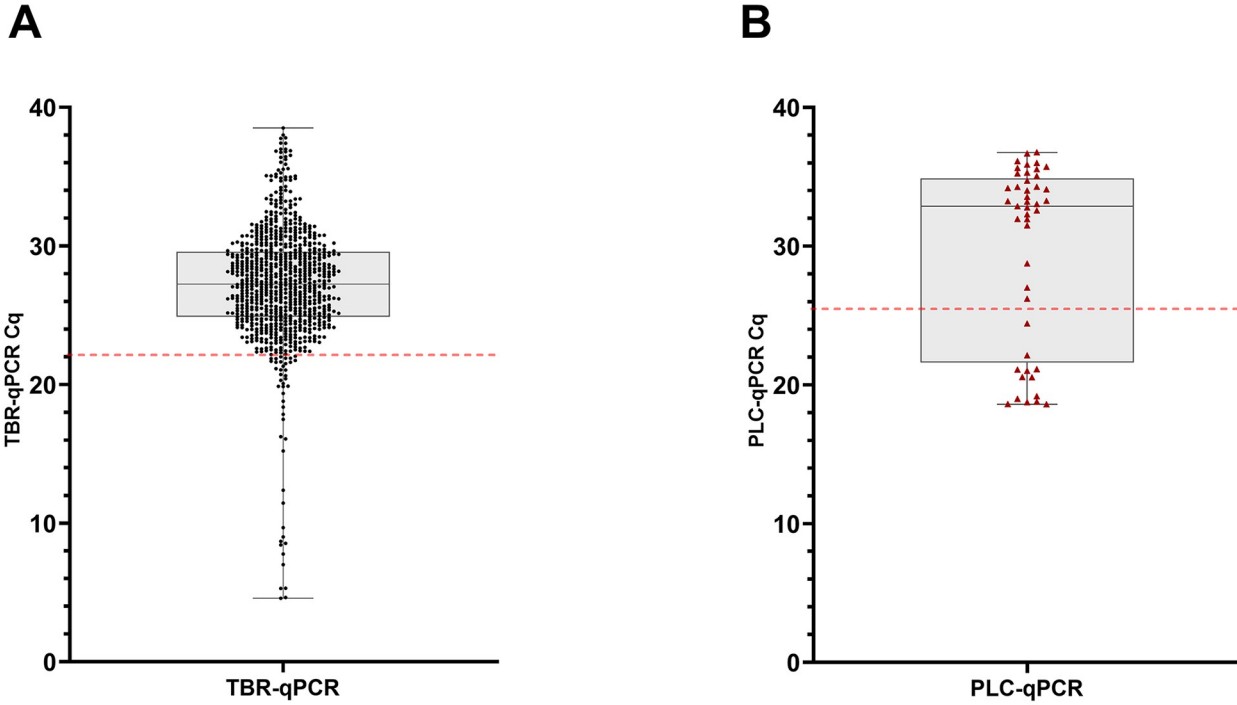

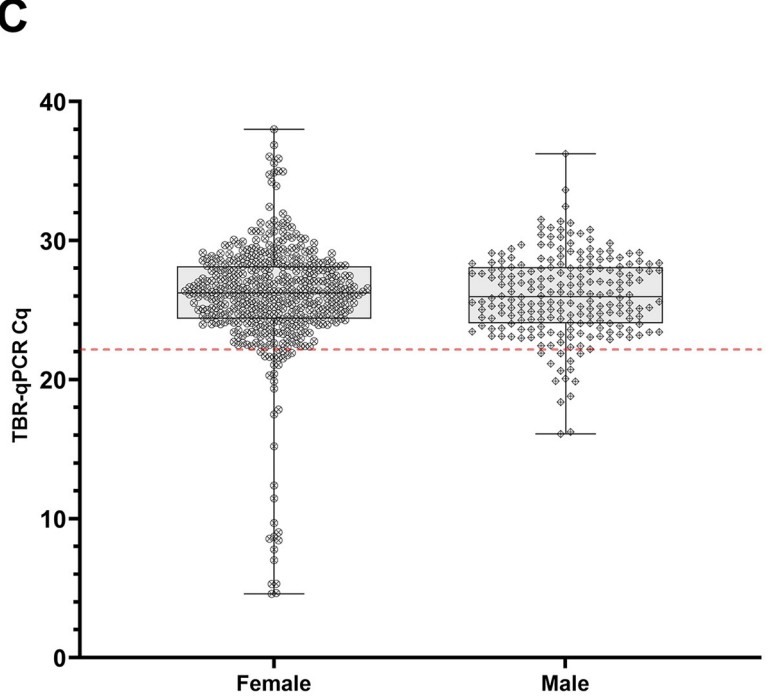

**Fig 4. Plots displaying Cq values for field-caught flies.** (A) shows TBR-qPCR Cq values (circular, black symbol) for all field flies where DNA was available (n = 640). (B) shows PLC-qPCR Cq values (triangular, red symbol) for a subset of field flies with TBR-qPCR Cq <22.13 and where DNA was available (n = 45). (C) shows comparison of TBR-qPCR Cq values from female (circular symbol, n = 428) and male (diamond symbol, n = 212) in field-caught flies. There was no significant difference in median TBR-qPCR Cq values from females (median = 26.22) and males (median = 25.97, $p$ = 0.5336). For all plots (A, B, C) grey boxplot shows median and 1–99% percentiles, error bars display range. The red dotted horizontal lines represent the Cq cut offs of 22.13 for TBR (A, C) and 25.36 for PLC (B).

**Table 3. A table displaying calculations of Cq cut-offs based on TBR-qPCR and PLC-qPCR screening of 45 infected flies (IFs), confirmed as midgut infection-positive by microscopy.**

| Assay | Mean (μ) Cq of IFs | SD (σ) of IFs | Cq Cut-off (μ + 3σ) | Lower CI (95%) | Upper CI (95%) |
|---|---|---|---|---|---|
| TBR-qPCR | 17.74 | 0.7458 | 22.13 | 21.56 | 22.70 |
| PLC-qPCR | 21.82 | 1.183 | 25.36 | 24.90 | 25.82 |

Given the data set is normally distributed, 99.7% of true IFs should lie within three standard deviations (SD, σ) of the mean (μ). CI = confidence interval.

population infection prevalence estimate was achieved in a two-step process using qPCR Cq cut-offs calculated from results of experiments with insectary-reared flies (Figs 3 and S3). Based on results from experimental, insectary-reared flies, a TBR-qPCR Cq cut-off of 22.13 (95% confidence interval (CI) [21.56, 22.70]) was determined for further analysis (Table 3). This was the mean TBR-qPCR Cq value of 45 insectary-reared IFs (17.74) added to three standard deviations (0.746). Any samples recording TBR-qPCR Cq values ≤ 22.13 were considered 'likely infected'. All flies in this subset were *G. pallidipes* (n = 45) and 71.1% (n = 32) were female. Furthermore, fly samples recording Cq values <16 were all female (n = 15; Fig 4).

Additional PLC screening was then carried out on this subset (TBR-qPCR Cq ≤ 22.13) of flies that had adequate volume of DNA available (n = 45), using a combination of HAT-HRM (n = 4) and PLC-qPCR (n = 41). All 45 samples recorded amplification when screened with PLC-qPCR, with Cq values ranging from 18.59 to 36.75 and mean of 29.71 ±0.968 SE (Fig 4). A PLC-qPCR cut-off of 25.36 (95% CI [24.90, 25.82]) was then calculated from the mean PLC-qPCR Cq value of 45 insectary-reared Ifs (21.82) added to three standard deviations (1.183; Table 3). Any samples recording PLC-qPCR Cq values ≤ 25.36 were considered 'true infected'. This left 13 individuals, all of which were female *G. pallidipes*. Sample population infection prevalence was therefore estimated to be 0.47% (13/2751) (95% CI [0.36, 0.73]), and *G. pallidipes* infection prevalence was estimated to be 0.79% (13/1650) (95% CI [0.61, 1.21]).

## Detection of *T. brucei* DNA by individual catch

There were 62 individual catches from which flies were collected and screened. Catches were from 35 different traps, across four transects (TA, TB, BA, BB) over seven discrete sampling days. A total of 24 catches (38.71%) contained at least 1 fly that tested positive by TBR-PCR. Of 62 catches, 19 met the analysis criteria of having >95% of flies collected and screened, and a total catch size of >1. When comparing the Cq values obtained from both TBR-qPCR and PLC-qPCR screening, it was apparent that across the 13 catches where *T. brucei* DNA was detected, six of the catches (BA9_13, BA5_15, TA5_01, TA1_01, BA8_13, BA3_15) contained one or two samples that recorded significantly lower Cq values (TBR-qPCR Cq 4.59–12.38, PLC-qPCR Cq 18.59–24.42) than other samples within the same catch (Fig 5). When using the respective Cq cut-offs for TBR-qPCR (22.13) and PLC-qPCR (25.36) to identify true infected samples (Table 3), it revealed infected flies were detected in five of these 19 catches (Fig 5), and eight of 62 total catches with a maximum of two infected flies per catch.

## Confirmation of *T. brucei* DNA in field samples

Sequencing of TBR-PCR 173 bp target products revealed high homology to *T. brucei* satellite DNA target entry (accession number K00392.1). Of 93 samples submitted, 91 returned sequences of suitable quality for BLAST analysis. Across forward and reverse sequences obtained from 91 different fly samples, BLAST analysis revealed and average percentage identity of 95.37% (± 0.137 SE). The variable homology is to be expected due to the heterogeneity of the target sequence [17]. A total of 69 of these sequences derived from TBR-PCR products

## A

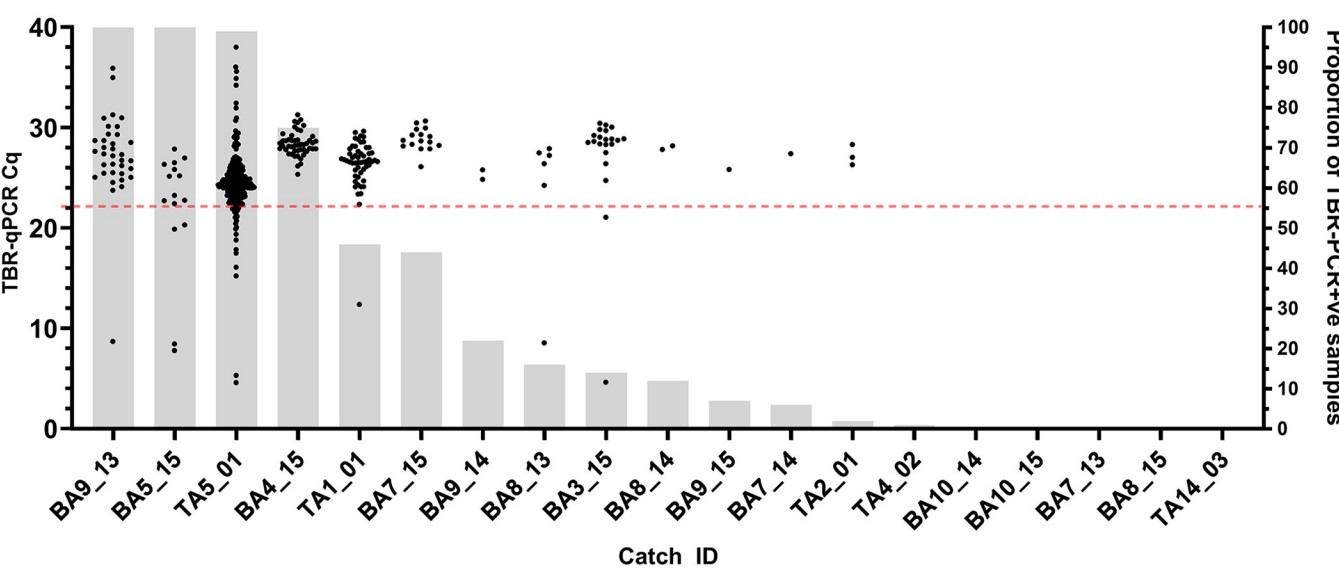

## B

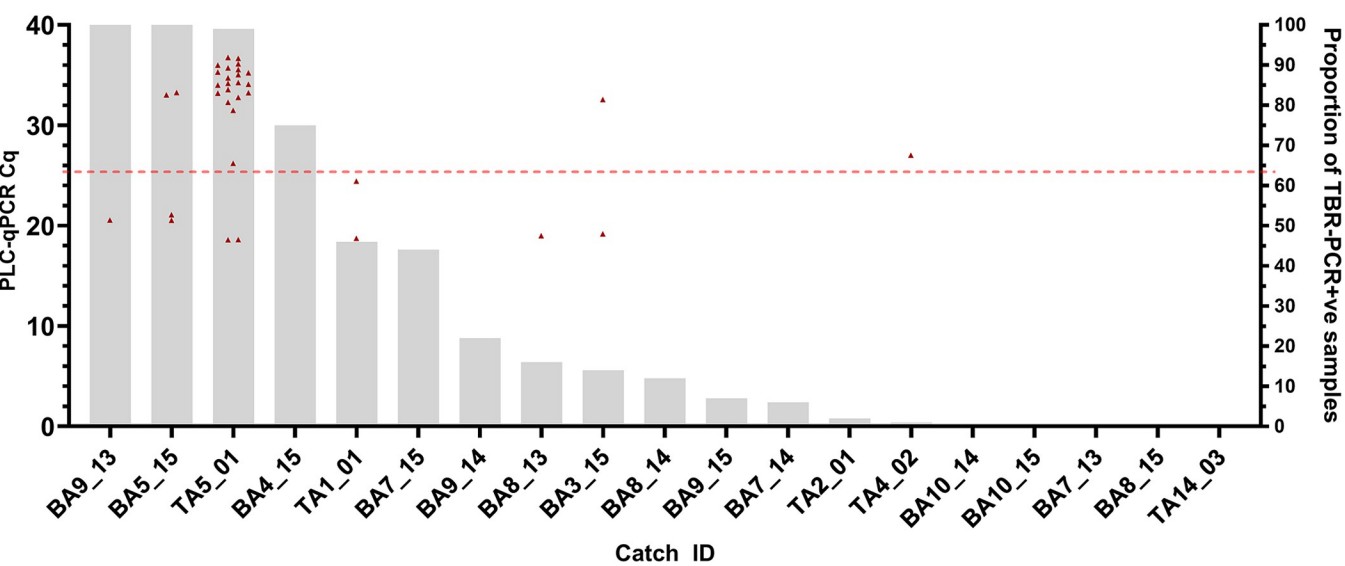

**Fig 5. Catches where >95% of trapped flies were collected and screened and total catch >1 (n = 19). Arranged in order of proportion of TBR +ve flies (L-R, largest to smallest).** (A) shows TBR-qPCR Cq values (circular, black symbol) for each fly sample in each catch. (B) shows PLC-qPCR Cq values (triangular, red symbol) for each fly sample in each catch that also had a TBR-qPCR Cq value <22.13 and had DNA available. Grey bars (right axes) represent the proportion (%) of flies in each catch testing TBR-PCR positive. The red dotted horizontal lines represent the Cq cut offs of 22.13 for TBR (A) and 25.36 for PLC (B).

were deposited in the National Center of Biotechnology Information (NCBI) GenBank database (accession numbers PP942745 to PP942812).

## Discussion

This study demonstrated that DNA from *T. brucei* infecting a tsetse can contaminate naïve uninfected tsetse within a trap cage environment, and that the level of contamination can be extensive. Even a low proportion of infected flies placed in a trap (1 infected:11 uninfected; T3) resulted in average 90.91% of uninfected flies producing a positive TBR-qPCR result (Fig 3) and 69% by TBR-PCR. Whilst the use of a less-sensitive assay (PLC-qPCR) led to a ten-fold reduction in false-negatives (T3; 9.1%), it did not remove the contamination effect entirely and still lead to false-positive results when used as an end-point assay. Conventional PCR and other DNA-based end-point assays (LAMP, RPA) that target *T. brucei* may therefore be highly sensitive, yet have insufficient specificity when used in xenomonitoring of *Glossina* sp. However, DNA quantification using quantitative PCR can help to eliminate false positive results. Our results showed clear demarcations in Cq value ranges between infected flies (true-positive) and contaminated naïve flies (false-positive) using both TBR-qPCR and PLC-qPCR (Fig 3). By considering Cq cut-offs, we were also able to determine that the proportion and quantity of *T. brucei* DNA contamination decreases with proportion of infected flies within the trap cage environment when co-housed for only 24 hours (Fig 3).

Low-level (Cq 32.21–36.64) localised air-borne contamination was also detected; using highly-sensitive TBR-qPCR, *T. brucei* DNA contamination (Cq < 40) was detected in 50% (n = 12/24) of negative control (naïve) flies in a trap cage (C0-A, C0-B) when placed in close proximity to cages housing infected flies (T1-3). This hypothesis was reinforced when there was no amplification of trypanosome DNA in control flies placed in a trap cage in a separate room (C0-C) (Fig 3). Aerosolised DNA contamination is a known phenomenon that can lead to false-positive results when screening for target DNA using PCR techniques [20,45]. Analytical sensitivity testing previously showed the TBR-qPCR assay as having a 95% limit-of-detection of 0.05–0.5 genomic equivalents per reaction [42]. These results highlight both the extreme sensitivity of the TBR genomic target and the care which should be taken when handling tsetse samples that may be infected with *T. brucei*. Detection of *T. brucei* DNA in tsetse, by either TBR-PCR or TBR-qPCR, is not indicative of a mature, transmissible infection. Consideration should be given to whether these assays are as biologically meaningful as dissections when used to estimate infection rate or prevalence, as concluded by Abdi *et al* [5].

Within-trap contamination was also evident in field samples. Using an end-point assay (TBR-PCR), a sample population was identified with a *T. brucei* DNA positivity rate of 24.77%. This far exceeds the expected infection prevalence in field flies [22]. Further to this, six catches recorded >40% infection prevalence by TBR-PCR, with three of these recording 99–100% proportion TBR-PCR positive (Fig 5). The largest of which (TA5_01) comprised 229 TBR-positive flies out of a possible 230 (Figs 5 and S4). As with the experimental insectary-reared flies, other potential sources of contamination, such as carry-over contamination during the DNA extraction or amplification stages, were ruled out by use of controls (negative extraction controls and negative template controls respectively). In addition, pre-amplification setup was performed within a dedicated PCR workstation with HEPA-filtered airflow, which is known to reduce aerosolised DNA contamination [46]. Using the quantitative DNA approach, a two-step Cq cut-off protocol revealed a more accurate true positive sample population infection prevalence of 0.47% (95% CI [0.36%, 0.73%]). This result is similar to that of a previous study conducted by Ngonyoka *et al* that reported a total *T. brucei* infection rate of 0.39% by ITS-PCR (a lower-sensitivity DNA target than TBR), in tsetse sampled from villages

also bordering Tarangire national park [47]. However, it is important to state that these results were also not validated by dissection. In the current study, all tsetse deemed to be likely infected were *G. pallidipes*, giving a *G. pallidipes* infection prevalence of 0.79%, although we do acknowledge the presence of species sampling bias across different transects (Table 2). This is slightly higher than the majority of *G. pallidipes* infection prevalences reported by previous studies (not using TBR-based methods) in Tanzania, which range from zero [48,49] to ~0.4% [47] but is lower than the 3.33% reported by Luziga *et al* [50]. Mature *T. brucei* infection in *G. pallidipes* is thought to be rare [22] as *G. pallidipes* are more refractory to trypanosome infection than *G. m. morsitans* [51].

The likely route of DNA contamination is *T. brucei* DNA in tsetse faeces from lysed or nonviable *T. brucei*. *Trypanosoma brucei* DNA has previously been detected in tsetse faecal material [32,33] and this was again confirmed in the present study (S3 Fig). Casual observations recorded during the laboratory-based experiments also noted high frequency of tsetse-tsetse interactions (mating and attempted mating) within trap cages in addition to defecation (S1H Fig). This agrees with previous research reporting that opportune male tsetse in particular will expend significant energy in seeking females repeatedly [52,53]. Bursell previously estimated that laboratory *G. m. morsitans* in 100% humidity conditions excreted approximately 30 μg of solid waste per hour at 76 hours after feeding [31]. Therefore, we would expect the defecation rate of the laboratory flies in the current study (72 hours post-feed), and hungry field flies, to have been similar. Faecal screening revealed that experimentally-infected tsetse consistently excreted *T. brucei* DNA from days 5 to 14 post-infection (S3 Fig). Flies that ingested an infected bloodmeal but did not have established infections (refractory flies) also excreted *T. brucei* DNA, with 32% (6/19) recording TBR-positive faecal samples and 74% (14/19) containing detectable TBR DNA at 20 days post-infection. However very low-level parasitaemia, undetected by microscopy, could account for this. The shedding of *T. brucei* DNA in the faeces of refractory flies demonstrates the possibility for trapped tsetse to contaminate their surroundings with *T. brucei* DNA without having established infections. It is important to state that we found no evidence to suggest that biological transmission can occur directly from tsetse to tsetse.

Fly parasitaemia is an important factor that likely influenced field results. In the current study we found that some field flies appear to contain much higher quantity of *T. brucei* DNA than the experimentally-infected *G. m. morsitans* flies. Whilst the minimum TBR-qPCR Cq value recorded for the experimentally infected flies was 15.09, field flies recorded Cq values as low as 4.59. All field flies recording Cq < 22.13 were *G. pallidipes*, and all field flies that recorded Cq <16 were female. Unfortunately, there is a paucity of studies quantifying parasitaemia or *Trypanosoma* DNA in either laboratory flies or, critically, field flies of any species. Possible explanations include older field flies (> 20 days) accumulating more parasites in the gut leading to higher parasitaemia, or simply the larger size of *G. pallidipes* [54] enabling them to ingest larger bloodmeals [34] and harbour more parasites.

The differences in parasitaemia between *G. pallidipes* sexes reported here agrees with previous field studies that have found higher rate of *Trypanosoma* infection in *G. pallidipes* females than males [48,55]. In addition, *G. pallidipes* females are larger than males with a 6.93% larger average wing length [34,54] and have been found to be more likely to develop mature infections than males, although not significantly more so [56]. Quantifying tsetse parasitaemia throughout infection stages in both the insectary and the field is an important next step in being able to determine more accurately infection rate or prevalence using molecular xenomonitoring methods. There are many biological factors that impact host-parasite interaction and parasitaemia in wild tsetse, and refinement of quantitative DNA cut offs may be required for different species and/or sexes.

Aside from parasitaemia, the quantity and proportion of *T. brucei* contamination modelled in the laboratory also does not necessarily apply in field catches. Although the large number of discarded flies prevented more in-depth analysis (S4 Fig), it was clear that for some catches the level of contamination was considerably greater than predicted, and in other cases less so (Fig 5). A variety of biological and environmental factors can influence DNA contamination in the field; catch size (1–420), fly density, higher average digestion rate (and thus potential defecation rate) in wild flies than in laboratory flies [57] and lack of decontamination measures between handling samples for sexing and morphological species identification. Conversely, there are factors in the field that may reduce DNA contamination, including DNA-degrading UV exposure, heat stress leading to adult fly morbidity [58], natural very low infection rates (< 3%) and the fact that not all flies would have been held in the trap cage for the maximum length of time (24 hours).

It is worth considering that in the current study DNA was extracted from whole tsetse flies, yet in several previous studies reporting high TBR positivity, DNA was extracted from dissected and excised tsetse midguts and/or salivary gland tissue only [24–26,28]. As we are hypothesising that *T. brucei* DNA contamination occurs in faecal samples on the fly exterior, contamination of internal tissues would only occur if they came into contact with the fly carapace during dissection. Whilst this is highly likely due to the nature of tsetse dissection, it is not assured.

It is not clear how the entomological trap and/or trap cage design impact contamination. In the current study, Nzi traps with plastic cage bottles were used for sampling Savanna tsetse in Tanzania. However, high prevalence of *T. brucei* s-l infection has also been reported in studies using Epsilon, biconical and pyramidal traps to capture a range of species in countries across West, Central and East Africa [23–27]. Although Musaya *et al* featured images of an epsilon trap with plastic bottle and biconical trap with transparent bag [26], the other studies do not detail the trap cage design. Methods to mitigate DNA contamination were not explored in this proof-of-principle study. Whilst fly density and refractoriness are beyond human control, changes can be made to the trap cage design and collection protocol to reduce tsetse mobilisation, tsetse-tsetse interaction and/or fly defecation. Another method is to wash samples prior to DNA extraction. Other arthropod studies have successfully used a sample-washing step to remove contaminant environmental DNA from species such as beetles, spiders and ticks [59–61]. For example, a 2.8% NaClO wash for 40 minutes at 7.5˚C, followed by three rounds of distilled water washes proved sufficient to remove contaminant DNA from the cuticles of individual wolf spiders (*Lycosidae*), yet did not appear to affect the quality of gut content DNA [60]. Although the quantitative methods described in the current study are a viable method for more accurately calculating infection rate where DNA contamination is suspected to have occurred, the focus should now be on measures to prevent or mitigate DNA contamination occurring in the first instance. The aim of this proof-of-principle study was to establish whether a viable DNA contamination route existed. Future research should therefore explore changes to trap cage design and sample-washing, keeping in mind field applicability, time and cost.

Molecular xenomonitoring is used in the surveillance of a range of other vector-borne diseases, some of which may also be susceptible to DNA contamination via vector faeces. DNA of *T. congolense* and *T. vivax*, causative agents of animal African trypanosomiasis, may also be shed in tsetse faeces. However, *Plasmodium falciparum*, *Wuchereria bancrofti* and *Mansonella perstans* DNA have all been detected in the excreta or faeces of *Anopheles* sp. [62]. This provides a viable pathway for DNA contamination within a mosquito trap. Whether the contamination does occur and to what extent it affects reporting of infection rates should be explored.

## Conclusions

During capture of infected tsetse, infected flies can passively contaminate uninfected ones with *T. brucei* DNA while they are retained in the cage of a trap both with insectary-reared and field caught tsetse. Although simple PCR may overestimate infection prevalence, qPCR offers a means of more accurately identifying parasite DNA in the tsetse. While these results can clearly impact tsetse surveillance, they may also have ramifications for xenomonitoring of other vector-borne diseases. Going forward, careful consideration should be given to vector trapping and collection methods in the molecular age. This could include DNA contamination, assay sensitivity and the way that results are interpreted. Future research should focus on methods to mitigate or eliminate DNA contamination within a trap cage and quantifying parasitaemia of mature salivary gland infection (confirmed vectors) in both laboratory and field-caught tsetse flies.

## Supporting information

**S1 Table. A table displaying *G. m. morsitans* sex and infection ratios for trap cage experiments. M = male, F = female, IF = infected fly, UF = naïve uninfected fly.**
(PDF)

**S2 Table. A table displaying transect, region and coordinates for each Nzi trap set as part of the study.** Tarangire NP = Tarangire National Park.
(PDF)

**S1 Fig. Images from experiments conducted on insectary-reared tsetse.** (a) and (b) show tsetse being blood-fed in solitary cells; (c) tsetse resting after bloodmeal; (d) tsetse solitary cells suspended above filter paper discs in rack; (e) collection of tsetse faecal samples on filter paper; (f) an infected tsetse marked with green oil paint; (g) experiment trap cages; (h) two infected tsetse (fly IDs 87 and 109) copulating inside trap cage during experiment; (i) dissected tsetse midgut infected with *T. brucei* as viewed under a microscope (400X).
(PDF)

**S2 Fig. Gel electrophoresis image from TBR-PCR screening of UFs in C0-A control trap (top row) and naïve flies in T1 control traps (bottom row).** Red arrow indicates target 173bp TBR product. NEC = negative extraction control. LAD = 100bp ladder.
(PDF)

**S3 Fig. Dissection and qPCR screening results of insectary-reared tsetse experimentally-infected with *Trypanosoma brucei brucei*.** S3A: A box-and-whisker plot (left axis) showing Cq values obtained from TBR-qPCR screening of faecal samples at four timepoints and eventual whole fly DNA from a subset of infected (IF) and refractory uninfected flies (UF) that were subject to dissection ante-mortem (n = 44). Samples from infected flies are in red, samples from refractory (uninfected) flies are in blue. The bars (right axis) shows the proportion of faecal samples recording TBR-qPCR amplification (where samples were available). The crosses represent the mean Cq values. The amount of *T. brucei* DNA detected in IF samples was consistently higher than that detected in UFs. Where amplification was recorded, there was a significant difference between mean TBR-qPCR Cq values from infected (mean = 17.57) and uninfected whole flies at 20 days (mean = 33.54, p = <0.0001). The midgut infection rate of this subset was 57% (25/44). S3B: Diagnostic positive predictive value (PPV) and negative predictive value (NPV) calculations for TBR-qPCR screening of tsetse faecal samples as a diagnosis of infection. Faecal samples collected 10–14 days post-inoculation that tested positive (TBR-qPCR) were highly likely to originate from an infected fly, with diagnostic positive

predictive value (PPV) of 91% and negative predictive value (NPV) of 85% A positive TBR-qPCR result ('qPCR_Y') was any sample recording amplification (Cq < 40). A negative TBR-qPCR result ('TBR-qPCR_N') was any sample that did not record amplification. Infected ('Infected_Y') was any fly confirmed as having mature midgut infection by microscopy, whilst uninfected ('Infected_N') was any fly confirmed as having no visible trypanosome infection by microscopy. Calculations are based on samples collected 10–12 days post-inoculation and/or 13–14 days post-inoculation.
(PDF)

**S4 Fig. Plots displaying total catch counts and respective sample TBR-qPCR Cq values for transects TA, TB and BA\*.** The left Y axis displays individual fly TBR-qPCR Cq values, plotted as black, circular symbols. The right Y axis displays number of flies caught in each catch, displayed as a stacked bar chart. Red shows the number of flies testing TBR-positive, blue shows the number of flies testing TBR negative, and grey shows the number of flies that were discarded and not collected. \*Transect BB is not featured, as it consisted of 1 TBR-negative fly caught in 1 trap (BB17_15).
(PDF)

## Acknowledgments

Special thanks go to Zachary Stavrou-Dowd and Dr Clair Rose for their invaluable assistance in tsetse colony maintenance at LSTM. Many thanks to Godfrey Mashenga, Kombo Shabani and Peter Daffa for their assistance and expertise in the trapping and collection of tsetse in Tanzania. Thanks also to Prof Wendy Gibson for providing *T. brucei* M249 DNA for use as positive control.

## Author Contributions

**Conceptualization:** Isabel Saldanha.

**Data curation:** Isabel Saldanha, Rachel Lea, Gala Garrod, Jennifer S. Lord.

**Formal analysis:** Isabel Saldanha.

**Funding acquisition:** Harriet Auty, Liam J. Morrison, Furaha Mramba, Stephen J. Torr.

**Investigation:** Isabel Saldanha, Rachel Lea, Oliver Manangwa, Gala Garrod, Stephen J. Torr.

**Methodology:** Isabel Saldanha, Lucas J. Cunningham.

**Project administration:** Rachel Lea, Harriet Auty, Jennifer S. Lord, Liam J. Morrison, Furaha Mramba, Stephen J. Torr.

**Resources:** Lee R. Haines, Álvaro Acosta-Serrano, Stephen J. Torr, Lucas J. Cunningham.

**Supervision:** Lee R. Haines, Álvaro Acosta-Serrano, Martha Betson, Stephen J. Torr, Lucas J. Cunningham.

**Validation:** Isabel Saldanha.

**Visualization:** Isabel Saldanha.

**Writing – original draft:** Isabel Saldanha.

**Writing – review & editing:** Isabel Saldanha, Rachel Lea, Lee R. Haines, Álvaro Acosta-Serrano, Martha Betson, Liam J. Morrison, Stephen J. Torr, Lucas J. Cunningham.

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
