## [Decision Letter · Decision Letter 0]

27 May 2024

Dear Ms Saldanha,

Thank you very much for submitting your manuscript "Caught in a trap: DNA contamination in tsetse xenomonitoring can lead to over-estimates of Trypanosoma brucei infection" for consideration at PLOS Neglected Tropical Diseases. As with all papers reviewed by the journal, your manuscript was reviewed by members of the editorial board and by several independent reviewers. The reviewers appreciated the attention to an important topic. Based on the reviews, we are likely to accept this manuscript for publication, providing that you modify the manuscript according to the review recommendations. 

Compliments to the authors on an interesting study that sheds light on the issue of cross-contamination in situations where tsetse flies are kept together in collection bottles for extended periods. The study is well executed and is of importance for studies on vector infections. 

The inclusion of a cleaning step to remove external contamination prior to processing for PCR, would have added value to understand the steps necessary to obtain really clean flies; for example, a washing step in water and bleach solution, or even a collection of flies directly into an Ethanol-based solution inside the trap bottles, could reduce/remove cross-contamination. 

I recommend that the authors include a statement explaining the possible inclusion of this step, but that their goal was to prove cross-contamination initially and that these cleansing steps could be investigated in future. 

Please also consider:

To deposit sequences generated to NCBI genbank.

Avoid starting sentences with abbreviations, eg. line 490.

Add references for arbovirus examples, Line 93-96.

Perhaps a better photograph of flies inside the collection bottle (Fig 2).

Sincerely,

Johan Esterhuizen

Guest Editor

Claudia Brodskyn

Section Editor

Compliments to the authors on an interesting study that sheds light on the issue of cross-contamination in situations where tsetse flies are kept together in collection bottles for extended periods. The study is well executed and is of importance for studies on vector infections. 

The inclusion of a cleaning step to remove external contamination prior to processing for PCR, would have added value to understand the steps necessary to obtain really clean flies; for example, a washing step in water and bleach solution, or even a collection of flies directly into an Ethanol-based solution inside the trap bottles, could reduce/remove cross-contamination. 

I recommend that the authors include a statement explaining the possible inclusion of this step, but that their goal was to prove cross-contamination initially and that these cleansing steps could be investigated in future. 

Please also consider:

To deposit sequences generated to NCBI genbank.

Avoid starting sentences with abbreviations, eg. line 490.

Add references for arbovirus examples, Line 93-96.

Perhaps a better photograph of flies inside the collection bottle (Fig 2).

Reviewer's Responses to Questions

**Key Review Criteria Required for Acceptance?**

**Methods**

-Are the objectives of the study clearly articulated with a clear testable hypothesis stated?

-Is the study design appropriate to address the stated objectives?

-Is the population clearly described and appropriate for the hypothesis being tested?

-Is the sample size sufficient to ensure adequate power to address the hypothesis being tested?

-Were correct statistical analysis used to support conclusions?

-Are there concerns about ethical or regulatory requirements being met?

Reviewer #1: The objectives are clearly defined and the study design is appropriate and well-executed. The experimental setup introduced sufficient sample size to be tested in control and field experiments

Reviewer #2: I compliment the authors on tackling an important question related to the novel surveillance and monitoring approaches for vector-borne disease and the production of interesting molecular approaches to limit cross-contamination and the bias in prevalence reporting from vectors.

However, my major criticism is that the authors do not report on the possibility of washing/cleaning the flies before molecular processing to limit these contaminations, it may be an easier way without the need for the whole battery of molecular test that may not very operational for large scale surveillance systems. More precisely:

134-136 given this risk of contamination why not wash the different flies before qPCR to prevent the detection of environmental contamination? Similarly, the quantitative methods displayed lines 447-452 are very good but again what about washing the samples before further processing is the contamination is only external.

Besides, I am totally convinced about how they relate the cage experiments in the lab with the trapping ones in the field, what about other trypanosome parasites and cross contamination in the field with thelleria, babesia or any others from biting flies that may also be trapped and so on.

Then I have one minor comment:

Lines 93-96, it would be good to add arboviruses references/examples.

Reviewer #3: The objectives of the study are clear, with a testable hypothesis. The study itself is well executed. The major shortcoming of this work is that they have excluded a clean-up stage in sample processing using established approaches (bleach and wash). This would have provided valuable information on whether end-point PCR is still valid as a protocol for xenomonitoring. The proposed qPCR is good, but not readily available in labs, especially where tropical infectious diseases are found.

The sample size is sufficient, and there are no concerns about ethics and regulatory issues.

**Results**

-Does the analysis presented match the analysis plan?

-Are the results clearly and completely presented?

-Are the figures (Tables, Images) of sufficient quality for clarity?

Reviewer #1: The analysis is thoroughly conducted and interpreted with sufficient graphs and statistical analysis

Reviewer #2: Good work on all the analyses

Reviewer #3: This section is well done, based on the samples and analysis undertaken. A few things to note:

Take note however that Figure 2 is poor, especially panel C that is unclear. Is this a fly on vegetation or in a trap?

Faecal samples of infected flies were analysed for the presence of trypanosomes (under Trap experiments). Why was it necessary to dissect the flies thereafter?

**Conclusions**

-Are the conclusions supported by the data presented?

-Are the limitations of analysis clearly described?

-Do the authors discuss how these data can be helpful to advance our understanding of the topic under study?

-Is public health relevance addressed?

Reviewer #1: The conclusions are supported by data with sufficient statistical analysis and suggestions for future studies. However, generated sequence data from TBR-PCR are not deposited to NCBI GenBank

Reviewer #2: No, the key point missing is the preprocessing of samples that could sort of void the need for the manuscript or instead increase the importance of it. The link between the lab and field part should also be made clearer.

Reviewer #3: This is an interesting study, because it can help to clarify shortcomings of xenomonitoring, and to provide opportunities to improve the approach. The very basic step of cleaning the surface of whole insects should be included.

**Editorial and Data Presentation Modifications?**

Reviewer #1: Accept with Minor Revision

Reviewer #2: (No Response)

Reviewer #3: None

**Summary and General Comments**

Reviewer #1: The authors conducted an important study detecting possible contamination and overestimation of trypanosome prevalence in vector flies. The study also demonstrated the sensitivity of the qPCR assays conducted however, the authors conducted TBR-PCR sequencing, and none of their sequences were deposited to NCBI GenBank for accessioning

The manuscript is novel and relevant to the journal as it highlights some of the errors that have been conducted in prevalence studies involving live flies as a means to detect parasitic infection stages within biological vectors 

It is recommended that the manuscript be accepted with minor revisions. The authors are encouraged to avoid starting sentences with abbreviations or numbers. Additionally, the authors are advised to deposit and submit all sequences generated.

Reviewer #2: I compliment the authors on tackling an important question related to the novel surveillance and monitoring approaches for vector-borne disease and the production of interesting molecular approaches to limit cross-contamination and the bias in prevalence reporting from vectors.

However, my major criticism is that the authors do not report on the possibility of washing/cleaning the flies before molecular processing to limit these contaminations, it may be an easier way without the need for the whole battery of molecular test that may not very operational for large scale surveillance systems. More precisely:

134-136 given this risk of contamination why not wash the different flies before qPCR to prevent the detection of environmental contamination? Similarly, the quantitative methods displayed lines 447-452 are very good but again what about washing the samples before further processing is the contamination is only external.

Besides, I am totally convinced about how they relate the cage experiments in the lab with the trapping ones in the field, what about other trypanosome parasites and cross contamination in the field with thelleria, babesia or any others from biting flies that may also be trapped and so on.

Then I have one minor comment:

Lines 93-96, it would be good to add arboviruses references/examples.

Reviewer #3: The study is very interesting. The authors have also published a paper describing finding T. brucei DNA and not T. Congolese in faecal material. Hence, this continues to add valuable information to understanding the complexities of epidemiological studies, especially in trypanosomiasis endemic country settings, often with limited capacity for molecular assays with high sensitivity that require enhance care in sample processing. However, the authors should include an assessment of whether DNA contamination remains, even after a good cleaning process. This may be difficult to do in the field, if infections are very low, and colony flies may be easier.

PLOS authors have the option to publish the peer review history of their article (what does this mean?). If published, this will include your full peer review and any attached files.

Reviewer #1: No

Reviewer #2: No

Reviewer #3: No

Figure Files:

Data Requirements:

Reproducibility:

References

---

## [Editor Report · Decision Letter 1]

26 Jul 2024

Dear Ms Saldanha,

We are pleased to inform you that your manuscript 'Caught in a trap: DNA contamination in tsetse xenomonitoring can lead to over-estimates of Trypanosoma brucei infection' has been provisionally accepted for publication in PLOS Neglected Tropical Diseases.

Best regards,

Johan Esterhuizen

Guest Editor

Guilherme Werneck

Section Editor

---

## [Editor Report · Acceptance letter]

7 Aug 2024

Dear Ms Saldanha,

We are delighted to inform you that your manuscript, "Caught in a trap: DNA contamination in tsetse xenomonitoring can lead to over-estimates of Trypanosoma brucei infection," has been formally accepted for publication in PLOS Neglected Tropical Diseases.

Best regards,

Shaden Kamhawi

co-Editor-in-Chief

Paul Brindley

co-Editor-in-Chief
